# INFERENCE & INTROSPECTION IN DEEP GENERATIVE MODELS OF SPARSE DATA

**Rahul G. Krishnan**
New York University
rahul@cs.nyu.edu

**Matthew Hoffman**
Adobe Research
matthoffm@adobe.com

## ABSTRACT

Deep generative models such as deep latent Gaussian models (DLGMs) are powerful and popular density estimators. However, they have been applied almost exclusively to dense data such as images; DLGMs are rarely applied to sparse, high-dimensional integer data such as word counts or product ratings. One reason is that the standard training procedures find poor local optima when applied to such data. We propose two techniques that alleviate this problem, significantly improving our ability to fit DLGMs to sparse, high-dimensional data. Having fit these models, we are faced with another challenge: how to use and interpret the representation that we have learned? To that end, we propose a method that extracts distributed representations of features via a simple linearization of the model.

## 1 INTRODUCTION

Deep latent Gaussian models (DLGMs, a.k.a. variational autoencoders; Rezende et al., 2014; Kingma et al., 2014) have led a resurgence in the use of deep generative models for density estimation. DLGMs assume that observed vectors $x$ are generated by applying a nonlinear transformation (defined by a neural network with parameters $\theta$) to a vector of Gaussian random variables $z$.

Learning in DLGMs proceeds by approximately maximizing the average marginal likelihood $p(x) \equiv \int_z p(z)p(x|z)dz$ of the observations $x$. Computing the true marginal likelihood is intractable, so we resort to variational expectation-maximization (Bishop, 2006), an approximation to maximum-likelihood estimation. To learn the parameters $\theta$ of the generative model, the procedure needs to find a distribution $q(z|x)$ that approximates the posterior distribution $p(z|x)$ of the latent vector $z$ given the observations $x$. In the past, such $q$ distributions were fit using iterative optimization procedures (e.g., Hoffman et al., 2013). But Rezende et al. (2014) and Kingma et al. (2014) showed that $q(z|x)$ can be parameterized by a feedforward "inference network" with parameters $\phi$, speeding up learning. This inference network is trained jointly with the generative model; as training proceeds, the inference network learns to approximate posterior inference on the generative model, and the generative model improves itself using the output of the inference network.

Embedded within this procedure, however, lies a potential problem: both the inference network and the generative model are initialized randomly. Early on in learning, the inference network's $q(z|x)$ distributions will be poor approximations to the true posterior $p(z|x)$, and the gradients used to update the parameters of the generative model will therefore be poor approximations to the gradients of the true log-likelihood $\log p(x)$. Previous stochastic variational inference methods (Hoffman et al., 2013) were slower, but suffered less from this problem since for every data-point, a set of variational parameters was optimized within the inner loop of learning. In this work, we investigate blending the two methodologies for learning models of sparse data. In particular, we use the parameters predicted by the inference network as an initialization and optimize them further during learning. When modeling high-dimensional sparse data, we show that updating the local variational parameters yields generative models with better held-out likelihood, particularly for deeper generative models.

What purpose is served by fitting bigger, deeper, more powerful generative models? Breiman (2001) argues that statistical discriminative modeling falls into two schools of thought: the data modeling culture and the algorithmic modeling culture. The former advocates the use of predictive models that assume interpretable, mechanistic processes while the latter advocates the use of black box techniques with an emphasis on prediction accuracy. Breiman's arguments also ring true about the

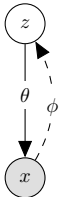

Figure 1: **Deep Latent Gaussian Model:** The Bayesian network depicted here comprises a single latent variable with the conditional probability $p(x|z)$ defined by a deep neural network with parameter $\theta$. The dotted line represents the inference network parameterized by $\phi$, which is used for posterior inference at train and test time.

divide between deep generative models with complex conditional distributions and simpler, more interpretable statistical models. Consider a classic model such as Latent Dirichlet Allocation (Blei et al., 2003). It is outperformed in held-out likelihood (Miao et al., 2016) by deeper generative models and assumes a simple probabilistic process for data generation that is unlikely to hold in reality. Yet, its generative semantics lend it a distinct advantage: interpretability. The word-topic matrix in the model allows practitioners to read off *what* the model has learned about the data. Is there a natural way to interpret the generative model when the conditional distributions are parameterized by a deep neural network?

Our second contribution is to introduce a simple, easy to implement method to interpret *what* is being learned by generative models such as DLGMs whose conditional probabilities are parameterized by deep neural networks. Our hope is to narrow the perceived gulf between a complex generative model's representational power and its interpretability. We use the Jacobian of the conditional distribution with respect to latent variables in the Bayesian network to form embeddings (or Jacobian vectors) of the observations. We investigate the properties of the Jacobian vectors obtained from deeper, more non-linear generative models.

## 2 BACKGROUND

**Generative Model:** We consider learning in generative models of the form shown in Figure 1. We observe a set of $D$ word count vectors $x_{1:D}$, where $x_{dv}$ denotes the number of times that word index $v \in \{1, \dots, V\}$ appears in document $d$. We assume we are given the total number of words per document $N_d \equiv \sum_v x_{dv}$, and that $x_d$ was generated via the following generative process:

$$z_d \sim \mathcal{N}(0, I); \ \gamma(z_d) \equiv \text{MLP}(z_d; \theta); \ \mu(z_d) \equiv \frac{\exp\{\gamma(z_d)\}}{\sum_v \exp\{\gamma(z_d)_v\}}; \ x_d \sim \text{Multinomial}(\mu(z_d), N_d). \tag{1}$$

That is, we draw a Gaussian random vector, pass it through a multilayer perceptron (MLP) with parameters $\theta$, pass the resulting vector through the softmax (a.k.a. multinomial logistic) function, and sample $N_d$ times from the resulting distribution over the vocabulary.[1]

**Variational Learning:** For ease of exposition notation we drop the subscript on $x_d$ to form $x$ referring to a single data point. We need to approximate the intractable posterior distribution $p(z|x)$ during learning. Using the well-known variational principle, we can obtain the lower bound on the log marginal likelihood of the data (or $\mathcal{L}(x; \theta, \phi)$) in Eq. 2. where the inequality is by Jensen's inequality.

$$\log p(x; \theta) \geq \mathbb{E}_{q_\phi(z|x)}[\log p_\theta(x|z))] - \text{KL}(\,q_\phi(z|x)||p(z)\,) = \mathcal{L}(x; \theta, \phi), \tag{2}$$

We leverage an *inference network* or *recognition network* (Hinton et al., 1995), a neural network which approximates the intractable posterior, during learning. This is a parametric conditional distribution that is optimized to perform inference. Kingma & Welling (2014); Rezende et al. (2014) use a neural net (with parameters $\phi$) to parameterize $q_\phi(z|x)$. The challenge in the resulting optimization problem is that the lower bound (2) includes an expectation w.r.t. $q_\phi(z|x)$, which implicitly depends on the network parameters $\phi$. This difficulty is overcome by using *stochastic backpropagation*.

With a normal distribution as our variational approximation we have that $q_\phi(z|x) \sim \mathcal{N}(\mu_\phi(x), \Sigma_\phi(x))$. $\mu_\phi(x), \Sigma_\phi(x)$ are functions of the observation $x$, and we denote by $\psi(x) :=$

---

[1] In keeping with common practice, we neglect the multinomial base measure term $\frac{N!}{x_1! \cdots x_V!}$, which amounts to assuming that the words are observed in a particular order.

$\{\mu_\phi(x), \Sigma_\phi(x)\}$ the local variational parameters predicted by the inference network. A simple transformation allows one to obtain unbiased Monte Carlo estimates of the gradients of $\mathbb{E}_{q_\phi(z|x)}[\log p_\theta(x|z))]$ with respect to $\phi$. If we assume the prior $p(z)$ is also normally distributed, the KL and its gradients may be obtained analytically. Throughout this paper we will use $\theta$ to denote the parameters of the generative model, and $\phi$ to denote the parameters of the inference network.

## 3 METHODOLOGY

**Inference with Global Information:** Sparse data typically exhibits long tails and learning in the presence of rare features is challenging. Inference networks learn to regress to the optimal posterior parameters for every data point and global information about the relative frequencies of the individual features in the training distribution may present valuable information during learning.

The simplest way to incorporate first order statistics across the training data into the inferential process is to condition on tf-idf (Baeza-Yates et al., 1999) features instead of the raw-counts. tf-idf is one of the most widely used techniques in information retrieval. In the context of building bag-of-words representations for documents, tf-idf re-weight features to increase the influence of rarer words while decreasing the influence of common words appearing in all documents. The tf-idf-transformed word-count vector is $\tilde{x}_{dv} \equiv x_{dv} \log \frac{D}{\sum_{d'} \min\{x_{d'v}, 1\}}$. After applying transform, the resulting vector $\tilde{x}$ is normalized by its L2 norm. It's worthwhile to note that leveraging first-order statistics for inference is difficult in the traditional paradigm of tracking variational parameters for each data point but is easy with inference networks.

**Optimizing Local Variational Parameters:** The inference network initially comprises a randomly initialized neural network. The predictions of the inference network early in optimization are suboptimal variational parameters used to derive gradients of the parameters of the generative model. This induces noise and bias to the gradients used to update the parameters of the generative model; this noise and bias may push the generative model towards a poor local optimum. Previous work has suggested that deep neural networks (which form the conditional probability distributions $p_\theta(x|z)$) are sensitive to initialization (Glorot & Bengio, 2010; Larochelle et al., 2009).

To avoid these issues, we only use the local variational parameters $\psi(x)$ predicted by the inference network to initialize an iterative optimizer that maximizes the ELBO with respect to $\psi$; we use the optimized variational parameters $\hat{\psi}(x)$ to derive gradients for the generative model. We then train the inference network using stochastic backpropagation and gradient descent, holding the parameters of the generative model $\theta$ fixed. Our procedure is detailed in Algorithm 1.

---

**Algorithm 1 Pseudocode for Learning:** We evaluate expectations in $\mathcal{L}(x)$ (see Eq. 2) using a single sample from the variational distribution and aggregate gradients across mini-batches. $M = 1$ corresponds to performing no additional optimization to the variational parameters We update $\theta, \psi(x), \phi$ using stochastic gradient descent with adaptive learning rates $\eta_\theta, \eta_{\psi(x)}, \eta_\phi$ obtained via ADAM (Kingma & Ba, 2015)

---

**Inputs**: Dataset $\mathcal{D} := [x_1, \ldots, x_D]$, Inference Model: $q_\phi(z|x)$, Generative Model: $p_\theta(x|z), p(z)$
**while notConverged**() **do**
 1. Sample datapoint: $x \sim \mathcal{D}$
 2. Estimate local variational parameters $\psi(x)_1$ using $q_\phi(z|x)$
 3. Estimate $\psi(x)_M \approx \hat{\psi}(x) = \arg\max_{\psi(x)} \mathcal{L}(x; \theta; \psi(x))$ via SGD as:
 $m = 1, \ldots, M, \psi(x)_{m+1} = \psi(x)_m + \eta_{\psi(x)} \frac{\partial \mathcal{L}(x; \theta, \psi(x)_m)}{\partial \psi(x)_m}$
 4. Update $\theta$ as: $\theta \leftarrow \theta + \eta_\theta \nabla_\theta \mathcal{L}(x; \theta, \psi(x)_M)$
 5. Update $\phi$ as: $\phi \leftarrow \phi + \eta_\phi \nabla_\phi \mathcal{L}(x; \theta, \psi(x))$
**end while**

---

**Introspection:** Linear models are inherently interpretable. Consider linear regression, factor analysis (Spearman, 1904), and latent Dirichlet allocation (LDA; Blei et al., 2003), which (standardizing notation) assume the following relationships:

$$\text{Regression: } \mathbb{E}[y|x] = Wx + b; \qquad \text{Factor Analysis: } x \sim \mathcal{N}(0, I); \quad \mathbb{E}[y|x] = Wx + b;$$
$$\text{Latent Dirichlet Allocation: } x \sim \text{Dirichlet}(\alpha); \quad \mathbb{E}[y|x] = Wx. \tag{3}$$

In each case, we need only inspect the parameter matrix $W$ to answer the question "what happens to $y$ if we increase $x_k$ a little?" The answer is clear—$y$ moves in the direction of the $k$th row of $W$. We can ask this question differently and get the same answer: "what is the derivative $\frac{\partial \mathbb{E}[y|x]}{\partial x}$?" The answer is simply the parameter matrix $W$.

For models as in Fig 1, the variability in the training data is assumed to be due to the single latent state $z$. The relationship between latent variables $z$ and observations $x$ cannot be quickly read off of the parameters $\theta$. But we can still ask what happens if we perturb $z$ by some small $dz$—this is simply the directional derivative $\frac{\partial \mathbb{E}[x|z]}{\partial z} dz$. We can interpret this Jacobian matrix in much the same way we would a factor loading matrix, with two main differences. First, the Jacobian matrix $\frac{\partial \mathbb{E}[x|z]}{\partial z}$ varies with $z$—the interpretation of $z$ may change significantly depending on context. Second, DLGMs exhibit rotational symmetry—the prior on $z$ is rotationally symmetric, and the MLP can apply arbitrary rotations to $z$ before applying any nonlinearities, so a priori there is no "natural" set of basis vectors for $z$. For a given Jacobian matrix, however, we can find the most significant directions via a singular value decomposition (SVD).

**Jacobian Vectors:** We present our method to generate embeddings from Bayesian networks of the form Figure 1. We consider three variants of Jacobian embedding vectors, based on the unnormalized potentials from the MLP, logarithmic probabilities, and linear probabilities respectively:

$$\mathcal{J}(z)^{\text{pot}} = \frac{\partial \gamma(z)}{\partial z} \qquad \mathcal{J}(z)^{\text{log}} = \frac{\partial \log \mu(z)}{\partial z} \qquad \mathcal{J}(z)^{\text{prob}} = \frac{\partial \mu(z)}{\partial z} \qquad (4)$$

For any $z$, $\{\mathcal{J}(z)^{\text{log}}, \mathcal{J}(z)^{\text{pot}}, \mathcal{J}(z)^{\text{prob}}\} \in \mathbb{R}^{V \times K}$ where $K$ is the latent dimension and $V$ is the dimensionality of the observations. It is this matrix that we use to form embeddings. We denote by $u_i$ the Jacobian vector obtained from the row of the Jacobian matrix. When not referring to a particular variant, we use $\mathcal{J}(z)$ to denote the Jacobian matrix. $\mathcal{J}(z)$ is a function of $z$ leaving open the choice of where to evaluate this function. The semantics of our generative model suggest a natural choice: $\mathcal{J}_{\textbf{mean}} := \mathbb{E}_{p(z)}[\mathcal{J}(z)]$. This set of embeddings captures the variation in the output distribution with respect to the latent state across the prior distribution of the generative model. Additionally, one may also evaluate the Jacobian at the approximate posterior corresponding to an observation $x$. We study how this may be used to obtain contextual word-vectors.

In frameworks that support automatic differentiation (e.g., Theano; Theano Development Team, 2016), $\mathcal{J}(z)$ is readily available and we estimate $\mathcal{J}_{\textbf{mean}}$ via Monte-Carlo sampling from the prior.

**Deriving Jacobian Vectors:** For simplicity, we derive the functional form of the Jacobian in a linear model i.e where $\gamma(z_d) = W z_d$ (c.f Eq 1). We drop the subscript $d$ and denote by $\gamma_i(z)$, the $i$th element of the vector $\gamma(z)$.

$$p(x_i = 1|z) = \frac{\exp(\gamma_i(z))}{\sum_j \exp(\gamma_j(z))} \qquad \text{and} \quad \gamma_i(z) = w_i^T z$$

For linear models, $\nabla_z \gamma_i(z) = w_i$ directly corresponds to $\mathcal{J}(z)^{\text{pot}}$. Noting that $\nabla_z \exp(\gamma_i(z)) = \exp(\gamma_i(z)) \nabla_z \gamma_i(z)$ and $\nabla_z \sum_j \exp(\gamma_j(z)) = \sum_j \exp(\gamma_j(z)) \nabla_z \gamma_j(z)$, we estimate $\mathcal{J}(z)^{\text{prob}}$ as:

$$\nabla_z p(x_i = 1|z) = \nabla_z \frac{\exp(\gamma_i(z))}{\sum_j \exp(\gamma_j(z))} = \frac{\sum_j \exp(\gamma_j(z)) \nabla_z \exp(\gamma_i(z)) - \exp(\gamma_i(z)) \nabla_z \sum_j \exp(\gamma_j(z))}{(\sum_j \exp(\gamma_j(z)))^2}$$

$$= \frac{\sum_j \exp(\gamma_j(z)) \exp(\gamma_i(z)) w_i - \exp(\gamma_i(z)) \sum_j \exp(\gamma_j(z)) w_j}{(\sum_j \exp(\gamma_j(z)))^2}$$

$$= p(x_i = 1|z) w_i - p(x_i = 1|z) \sum_j p(x_j = 1|z) w_j$$

$$= p(x_i = 1|z)(w_i - \sum_j p(x_j = 1|z) w_j)$$

Similarly, we may compute $\mathcal{J}(z)^{\text{log}}$:

$$\nabla_z \log p(x_i = 1|z) = w_i - \sum_j p(x_j = 1|z) w_j = \sum_j p(x_j = 1|z)(w_i - w_j) \qquad (5)$$

Denoting a word-pair vector as $w_i - w_j$, where $w_i, w_j$ are columns of the matrix $W$. If we define the set of all word-pair vectors as $\mathcal{S}$, then Eq 5 captures the idea that the vector representation for a word $i$ lies in the convex hull of $\mathcal{S}$. Furthermore, the word vector's location in $\text{CONV}(\mathcal{S})$ is determined by the likelihood of the pairing word $(x_j)$ under the model $p(x_j = 1|z)$.

When we use a non-linear conditional probability distribution $\mathcal{J}(z)^{\log}$ becomes: $\nabla_z \log p(x_i = 1|z) = \sum_j p(x_j = 1|z)(\nabla_z \gamma_i(z) - \nabla_z \gamma_j(z))$ where $\nabla_z \gamma_i(z)$ is a non-linear function of $z$. To the best of our knowledge, Jacobian Vectors and their properties have not been studied.

## 4 RELATED WORK

*Learning in Deep Generative Models:* Salakhutdinov & Larochelle (2010) optimize the local variational parameters obtained from an inference network when learning deep Boltzmann machines. For DLGMs, Hjelm et al. (2016) also consider the optimization of the local variational parameters, though their exposition focuses on deriving an importance-sampling-based bound to use during learning in deep generative models with discrete latent variables. Their experimental results suggest the procedure does not improve performance much on the binarized MNIST dataset. This is consistent with our experience—we found that our secondary optimization procedure helped more when modeling sparse, high-dimensional text data than when modeling MNIST.

*Leveraging Gradient Information:* The algorithmic procedure for obtaining Jacobian Vectors that we propose resembles that used to derive Fisher Score features. For an data point $X$ under a parameteric distribution $p(X; \theta)$, the Fisher scores is defined as $U_X = \nabla_\theta \log p(X; \theta)$. Jaakkola & Haussler (2007) similarly use $U_X$ to form a kernel function for subsequent use in a discriminative classifier. The intuition behind such methods is to note that the derivative of the log-probability with respect to the parameters of the generative model encodes all the variability in the input under the generative process. We rely on a related intuition, although our motivations are different; we are interested in characterizing isolated features such as words, not vector observations such as documents. Also, we consider Jacobians with respect to per-observation latent variables $z$, rather than globally shared parameters $\theta$.

In the context of discriminative modeling, (Erhan et al., 2009) use gradient information to study the patterns with which neurons are activated in a deep neural networks while (Wang et al., 2016) use the spectra of the Jacobian to study the complexity of the functions learned by neural networks.

*Introspection via Embeddings:* Landauer et al. (1998) proposed latent semantic analysis, one of the earliest works to create vector space representations of documents. Bengio et al. (2003); Mikolov & Dean (2016) propose log-linear models to create word-representations from document corpora in an unsupervised fashion. Rudolph et al. (2016) describe a family of models to create contextual embeddings where the conditional distributions that lie in the exponential family. Finally, Choi et al. (2016) propose a variant of Word2Vec to create representations of diagnosis codes from temporal Electronic Health Record data. The models above explicitly condition the probability of a word on its nearby context. In contrast, our model models the probability of a word as it appears in the document (or rather, conditioned on its global context). Augmenting the generative model in Figure 1 to incorporate local context is a possible direction for future work.

Miao et al. (2016) learn a shallow log-linear model on text data and obtain embeddings for words from the weight matrix that parameterize their generative model. Li et al. (2016) propose a modification to LDA that explicitly models representations for word in addition to modeling the word-topic structure.

## 5 EVALUATION

*Text Data:* We study the effect of further optimization of the variational parameters and inference with tf-idf features on the two datasets of varying size: the smaller 20Newsgroups (Lang, 2008) (train/valid/test: 10768/500/7505, $V$: 2000) and the larger RCV2 (Lewis et al., 2004) dataset (train/valid/test: 789414/5000/10000, $V$: 10000). We follow the preprocessing procedure defined in (Miao et al., 2016) for both datasets. We also train models on the Wikipedia corpus used in (Huang et al., 2012). We remove stop words, words appearing less than ten times in the dataset. and select our vocabulary to comprise the union of the top 20000 words in the corpus, the words in the WordSim353

(Finkelstein et al., 2001) and the words in the Stanford Contextual Word Similarity Dataset (SCWS) (Huang et al., 2012). The resulting dataset is of size train/valid: $1212781/1000$ and $V$:20253.

*EHR Data:* We train shallow and deep generative models on a dataset constructed from Electronic Medical Records. The dataset comprises $185000$ patients where each patient's data across time was aggregated to create a bag-of-diagnosis-codes representation of the patient. The vocabulary comprises four different kinds of medical diagnosis codes: ICD9 (diagnosis), LOINC (laboratory tests), NDC (prescription medication), CPT (procedures). For a single patient, we have $51321$ diagnosis codes.

**Training Procedure:** On all datasets, we train shallow log-linear models ($\gamma(z) = Wz + b$) and deeper three-layer DLGMs ($\gamma(z) = \text{MLP}(z; \theta)$). We vary the number of secondary optimization steps $M = 1, 100$ (cf. Algorithm 1) to study the effect of optimization on $\psi(x)$ with ADAM (Kingma & Ba, 2015). We use a mini-batch size of $500$, a learning rate of $0.01$ for $\psi(x)$ and $0.0008$ for $\theta, \phi$. The inference network was fixed to a two-layer MLP whose intermediate hidden layer $h(x)$ was used to parameterize the mean and diagonal log-variance $\mu(x), \log \Sigma(x)$. To evaluate the quality of the learned generative models, we report an upper bound on perplexity (Mnih & Gregor, 2014) given by $\exp(-\frac{1}{N} \sum_i \frac{1}{N_i} \log p(x_i))$ where $\log p(x_i)$ is replaced by Eq 2. The notation 3-M100-tfidf indicates a model where the MLP parameterizing $\gamma(z)$ has three hidden layers, the local variational parameters are updated 100 times before an update of $\theta$ and tf-idf features were used in the inference network.

**Improving Learning:** Table 1 depicts our results on 20newsgroups and RCV2. On the smaller dataset, we find that the deeper models overfit quickly and are outperformed by shallow generative models. On the larger datasets, the deeper models' capacity is more readily utilized yielding better generalization. The use of tf-idf features always helps learning on smaller datasets. On larger datasets, the benefits are smaller when we also optimize $\psi(x)$. Finally, the optimization of the local variational parameters appears to help most on the larger datasets. To investigate how this occurs, we plot the held-out likelihood versus epochs. For models trained on the larger RCV2 (Figure 2a) and Wikipedia (Figure 2b) datasets, the larger deep generative models converge to better solutions (and in fewer passes through the data) with the additional optimization of $\psi(x)$.

To study the effect of investigate where optimizing $\psi(x)$ is particularly effective on, we train a three layer model on different subsets of the Wikipedia dataset. The subsets are created by selecting the top $K$ most frequently occurring features in the data. Our rationale is that by holding everything fixed and varying the level of sparsity (datasets with smaller values of $K$ are less sparse) in the data, we can begin to understand when our method is most helpful. On held-out data, we compute the difference between the perplexity when the model is trained with $M = 1$ (denoted $P_{M1}$) and $M = 100$ (denoted $P_{M100}$) and compute the relative decrease in perplexity obtained as $\frac{P_{M1} - P_{M100}}{P_{M100}}$. The results are depicted in Figure 2c where we see that our method improves learning as a function of the dimensionality of the data.

In Table 5 in the supplementary material, we study the effect of varying the parameters of the inference network. There, we perform a small grid search over the hidden dimension and the number of layers in the inference network and find that optimizing the variational parameters continues to produces models with lower overall perplexity.

**Jacobian Vectors:** Our first avenue for introspection into the learned generative model is using log-singular values of the Jacobian matrix. Since the Jacobian matrix precisely encodes how sensitive the outputs are with respect to the inputs, the log-singular value spectrum of this matrix directly captures the amount of variance in the data explained by the latent space. Said differently, we can read off the number of active units in the DLGM or VAE by counting the number of log-singular values larger than zero. Furthermore, this method of introspection depends only on the parameters of the generative model. In Figure 2d, 2e, we see that for larger models continuing to optimize the variational parameters allows us to learn models that use many more of the available latent dimensions. This suggests that, when fit to text data, DLGMs may be particularly susceptible to the overpruning phenomenon noted by Burda et al. (2015). In Figure 2, the lower held-out perplexity and the increased utilization of the latent space suggest that the continued optimization of the variational parameters yields more powerful generative models.

We investigate how the Jacobian matrix may be used for model introspection by studying the qualitative properties of $\mathcal{J}_{\text{mean}}^{\log}$ on DLGMs (of type "3-M100-tfidf") trained on two, diverse sets of data. We form a Monte Carlo estimate of $\mathcal{J}_{\text{mean}}^{\log}$ using $400$ samples. The cosine distance is

Table 1: **Test Perplexity: Left: Baselines** Results on the 20newsgroups and RCV1-v2 dataset Legend: LDA (Blei et al., 2003), Replicated Softmax (RSM) (Hinton & Salakhutdinov, 2009), Sigmoid Belief Networks (SBN) and Deep Autoregressive Networks (DARN) (Mnih & Gregor, 2014), Neural Variational Document Model (Miao et al., 2016). $K$ denotes the latent dimension in our notation. **Right:** DLGMs on text data with $K = 100$. We vary the features presented to the inference network $q_\phi(z|x)$ during learning between: normalized count vectors ($\frac{x}{\sum_{i=1}^V x_i}$, denoted "norm") and normalized tf-idf (denoted "tf-idf") features.

| Model | $K$ | 20News | RCV1-v2 |
|---|---|---|---|
| LDA | 50 | 1091 | 1437 |
| LDA | 200 | 1058 | 1142 |
| RSM | 50 | 953 | 988 |
| SBN | 50 | 909 | 784 |
| fDARN | 50 | 917 | 724 |
| fDARN | 200 | — | 598 |
| NVDM | 50 | 836 | 563 |
| NVDM | 200 | 852 | 550 |

| DLGM | 20News | | RCV1-v2 | |
|---|---|---|---|---|
| | M1 | M100 | M1 | M11 |
| 1-M1-norm | 964 | 816 | 498 | 479 |
| 1-M100-norm | 1182 | 831 | 485 | 453 |
| 3-M1-norm | 1040 | 866 | 408 | 360 |
| 3-M100-norm | 1341 | 894 | 378 | 329 |
| 1-M1-tfidf | 895 | **785** | 475 | 453 |
| 1-M100-tfidf | 917 | 792 | 480 | 451 |
| 3-M1-tfidf | 1027 | 852 | 391 | 346 |
| 3-M100-tfidf | 1029 | 833 | 377 | **327** |

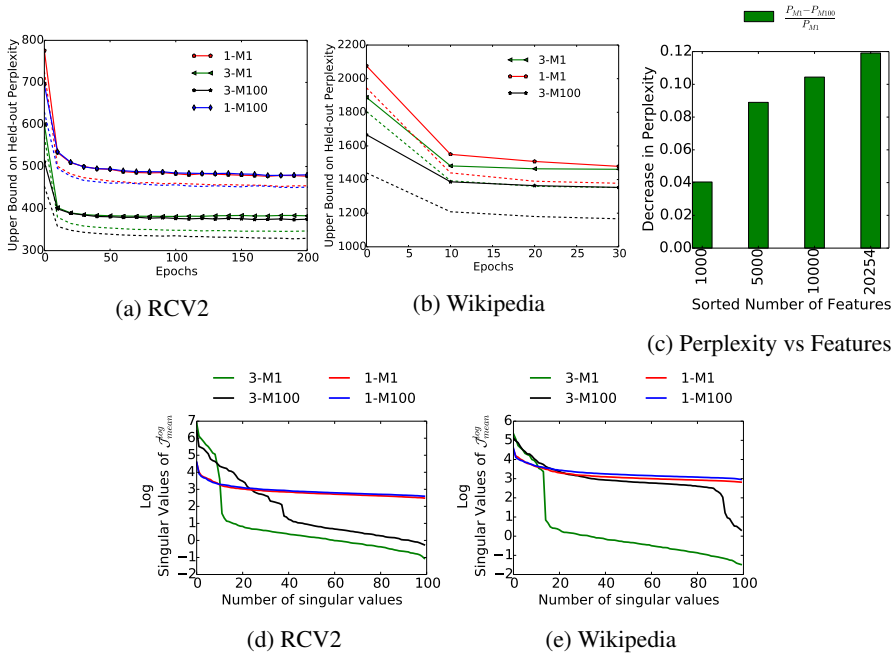

(a) RCV2  (b) Wikipedia

(c) Perplexity vs Features

(d) RCV2  (e) Wikipedia

Figure 2: **Mechanics of Learning:** *Validation Perplexity and Log-singular Values of $\mathcal{J}_{mean}^{log}$:* Best viewed in color. For the RCV2 and Wikipedia (large) datasets, we visualize the validation perplexity as a function of epochs. The solid lines indicate the validation perplexity for $M = 1$ and the dotted lines indicate $M = 100$. The x-axis is **not** directly comparable on running times since larger values of $M$ take longer during training. We find that learning with $M = 100$ takes approximately 15 times as long per mini-batch of size 500 on the text datasets. Figure 2c compares relative differences in the final held-out perplexity, denoted $P$, between models trained using $M = 1$ and $M = 100$. On the x-axis, we vary the number of features used in the dataset. Figure 2d, 2e depict the sorted log singular values of $\mathcal{J}_{mean}^{log}$.

used to define neighbors of words in the embedding space of the Jacobian and spectral clustering (Von Luxburg, 2007) is used to form clusters.

In Table 2a, we visualize some of the nearest neighbors of words using $\mathcal{J}_{mean}^{log}$ obtained from models trained on the Wikipedia dataset. The neighbors are semantically sensible. Instead of evaluating the Jacobian at $L$ points $z_{1:L} \sim p(z)$, one may instead evaluate it at $z_{1:L} \sim q(z|x)$ for some $x$. In Table 2b, we select three polysemous query words alongside "context words" that disambiguate the query's meaning. For each word-context pair, we create a document comprising a subset of words in the the

(a) **Word Embeddings (Nearest Neighbors):** We visualize nearest neighbors of word embeddings. We exclude plurals of the query and other words in the neighborhood.

| Query | Neighborhood |
|---|---|
| intelligence | espionage, secrecy, interrogation, counterterrorism |
| zen | dharma, buddhism, buddhas, meditation, yoga |
| artificial | artificially, molecules, synthetic, soluble |
| military | civilian, armys, commanders, infantry |

(b) **Word Embeddings (Polysemy):** We visualize the nearest neighbors under the Jacobian vector induced by the posterior distribution of a document created based on the context word.

| Word | Context | Neighboring Words |
|---|---|---|
| crane | construction | lifting, usaaf, spanned, crushed, lift |
|  | bird | erected, parkland, locally, farmland, causeway |
| bank | river | watershed, footpath, confluence, drains, tributary |
|  | money | banking, government, bankers, comptroller, fiscal |
| fires | burn | ignition, combustion, engines, fuel, engine |
|  | layoff | thunderstorm, grassy, surrounded, walkway, burning |

(c) **Medical Embeddings (Nearest Neighbors):** We evaluate nearest neighbors of selected diagnosis, drug and procedure codes (ignoring duplicates and shortening some code names). Metformin, Glimepiride, Pioglitazone and Avandia are diabetic drugs. A contour meter is an instrument to track blood glucose. Advair, Albuterol, Proventil and Spiriva are prescribed to patients with chronic obstructive pulmonary disease (COPD)

| Code | Neighboring Codes |
|---|---|
| Metformin | Glimepiride, Avandia, Contour Meter, Pioglitazone |
| Spiriva (Bronchodilator) | Advair , Albuterol , Proventil |
| Asbestosis | Exposure To Asbestos , Coal Workers' Pneumoconiosis, Ct Scan Chest |

Table 2: **Qualitative evaluation of Jacobian Vectors :** In Table 2a and 2b, we evaluate the embeddings of words. In Table 8b and 2c we evaluate embeddings of medical diagnosis codes.

| Models | Spearman $\rho \times 100$ | Models | Spearman $\rho \times 100$ |
|---|---|---|---|
| Huang(G) | 22.8 | Huang (S) | 58.6 |
| Huang | 71.3 | Huang (M) | 65.7 |
| Glove | 75.9 | C&W | 57 |
| C&W | 55.3 | tf-idf-S | 26.3 |
| ESA | 75 | Pruned tf-idf-S | 62.5 |
| Tiered Pruned tf-idf | 76.9 | Pruned tf-idf-M | 60.5 |
| 1-M1 $\mathcal{J}_{\mathbf{mean}}^{\mathrm{prob}}$ | 69.7 | 1-M1 $\mathcal{J}_{\mathbf{mean}}^{\mathrm{prob}}$ | 61.7 |
| 3-M100 $\mathcal{J}_{\mathbf{mean}}^{\mathrm{prob}}$ | 59.6 | 3-M100 $\mathcal{J}_{\mathbf{mean}}^{\mathrm{prob}}$ | 59.5 |

(a) **WordSim353:** "G" denotes the model in Huang et al. learned only with global context in the document.

(b) **SCWS:** (S) denotes a single prototype approach versus (M) that denotes a multi-prototype approach (that leverages context)

Table 3: **Semantic Similarity in Words:** The baseline results are taken from (Huang et al., 2012). C&W uses embeddings from the language model of (Collobert & Weston, 2008). Glove corresponds to embeddings by (Pennington et al., 2014). The learning algorithm for our embeddings does not use local context.

context's Wikipedia page. Then, we use the learned inference network to perform posterior inference to evaluate $\mathcal{J}_{\mathbf{mean}}^{\mathrm{log}}$ at the corresponding $q(z|x)$. This yields a set of contextual Jacobian vectors. We display the nearest neighbors for each word under different contextual Jacobian vectors and find that, while not always perfect, they capture *different* contextually relevant semantics. The take-away here is that by combining posterior inference in this Bayesian network with our methodology of introspecting the model, one obtains *different* context-specific representations for the observations despite not having been trained to capture this explicitly.

In Table 8b (appendix), we visualize clusters formed from the embeddings of medical diagnosis codes to find that they exhibit topical coherence. In Table 2c, the nearest neighbors of drugs include other drugs prescribed in conjunction with or as a replacement for the query drug. For diagnosis codes such as "Asbestosis", the nearest neighbors are symptoms and procedures associated with the

Table 4: **Medical Relatedness Measure:** Evaluating the quality of embedding using medical (NDF-RT and CCS) ontologies. Each column corresponds to a measure of how well the embedding space is amenable to performing analogical reasoning (NDF-RT) or clusters meaningfully (CCS). A higher number is better. SCUIs corresponds to the application of the method developed by (Choi et al., 2016) on data released by (Finlayson et al., 2014). The learning algorithm for our embeddings does not use local context.

| Models | $\text{MRM}_{\text{NDF-RT}}$ (May Treat) | $\text{MRM}_{\text{NDF-RT}}$ (May Prevent) | $\text{MRM}_{\text{CCS}}$ (Fine Grained) | $\text{MRM}_{\text{CCS}}$ (Coarse Grained) |
|---|---|---|---|---|
| (De Vine et al., 2014) | 53.21 | 57.14 | 22.63 | 24.56 |
| (Choi et al., 2016) | 59.40 | 55.71 | 44.80 | 47.43 |
| SCUI | 52.75 | 48.57 | 34.16 | 37.31 |
| 1-M1 $\mathcal{J}_{\text{mean}}^{\text{pot}}$ | 59.63 | 32.86 | 31.58 | 33.88 |
| 3-M100 $\mathcal{J}_{\text{mean}}^{\text{pot}}$ | 60.32 | 38.57 | 37.77 | 40.87 |

disease. Finally, for a qualitative evaluation of Jacobian vectors obtained from a model trained on movie ratings, we refer the reader to the appendix.

**The Semantics of Embeddings:** We evaluate the vector space representations that we obtain from $\mathcal{J}_{\text{mean}}$ on benchmarks (such as WordSim353 (Finkelstein et al., 2001) and SCWS (Huang et al., 2012)) that attempt to measure the similarity of words. The algorithmically derived measure of similarity is compared to a human-annotated score (between one and ten) using the Spearman rank correlation. The models that we compare to primarily use local context, which yields a more precise signal about the meanings of particular words. Closest to us in terms of training procedure is (Huang (G)) in Table 3a, whose model we outperform. Finding ways to incorporate local context is fertile ground for future work on models tailor-made for extracting embeddings.

For medical codes, we follow the method in (Choi et al., 2016). The authors build two kinds of evaluations to estimate whether an embedding space of medical diagnosis codes captures medically related concepts well. $\text{MRM}_{\text{NDF-RT}}$ (Medical Relatedness Measure under NDF-RT) leverages a database (NDF-RT) to evaluate how good an embedding space is at answering analogical queries between drugs and diseases such as $u_{\text{Diabetes}} \approx u_{\text{Metformin}} - (u_{\text{Lung Cancer}} - u_{\text{Tarceva}})$. (Metformin is a diabetic drug and Tarceva is used in the treatment of lung cancer). The evaluation ($\text{MRM}_{\text{CCS}}$) measures if the neighborhood of the diagnosis codes is medically coherent using a predefined medical ontology (CCS) as ground truth. The number computed may be thought of as a measure of precision, where a higher number is better. We refer the reader to the appendix for additional details.

Table 4 details the results on evaluating the medical embeddings. Once again, the baselines we compare (Choi et al., 2016) are variants of Word2Vec that maximize the likelihood of the diagnosis codes conditioned on carefully crafted contexts. Our method performs comparably to the baselines, even though it relies exclusively on global context and was not designed with this task in mind. This setting depicts an instance where Jacobian vectors resulting from a deeper, better-trained model outperform those from a shallow model, highlighting the importance of a method of interpretation agnostic to the structure of the conditional probability functions in the generative model.

Between the three choices of Jacobian vectors, we found that all three perform comparably on the word similarity evaluation with $\mathcal{J}_{\text{mean}}^{\text{prob}}$ slightly outperforming the others. On the medical data, with we found similar results aside from a few cases where $\mathcal{J}_{\text{mean}}^{\text{prob}}$ did not perform well. For deeper models, we found that optimizing $\psi(x)$ improved the quality of the obtained Jacobian vectors on text and medical data. The full versions of Tables 3 and 4 can be found in the appendix.

## 6 DISCUSSION

We explored techniques to improve inference and learning in deep generative models of sparse non-negative data. We also developed and explored a novel, simple, yet effective method to interpret the structure of the non-linear generative model via embeddings obtained from the Jacobian matrix relating latent variables to observations. The embeddings are evaluated qualitatively and quantitatively, and were seen to exhibit interesting semantic structure across a variety of domains. Studying the effects of varying the priors on the latent variables, conditioning on context, and varying the neural architectures that parameterize the conditional distributions suggest avenues for blending ideas from generative modeling and Bayesian inference into building more powerful embeddings for data.

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

## APPENDIX A    RCV2: COMPLEXITY OF THE INFERENCE NETWORK

We study the effect of varying the inference network on the two-layer model trained with the RCV2 data. Folding fixed a three-layer DLGM with stochastic dimension $100$ (same architecture as $3-tfidf$ in Table 1), we learn models on the RCV2 data using $M = 1, 100$ (we evaluate and report bounds on perplexity using $M = 100$) and display the results in Table 5. When $M = 1$ (the standard procedure for training VAEs and DLGMs), increasing the number of layers in the inference network decreases the quality of the model learned. One possible explanation for this is that the already noisy gradients of the inference network must propagate along a longer path in a deeper inference network, slowing down learning of the parameters $\phi$ which in turn affects the quality of inference. In contrast, increasing hidden dimension of the inference network improves results. Generally, we obtain better results (in both train and validation error) with $M = 100$ than training with $M = 1$ across the various configurations of the inference network that we tried. Furthermore, we find that when $M = 100$, the inference network architecture is less relevant and all models converge to approximately the same result suggesting that the procedure treats the output of the inference network as a crude initialization for the variational parameters and that the subsequent steps of optimization are primarily responsible for gains in learning.

Table 5: **Train and Test Perplexity on RCV2:**   The tables herein show the train and held out bounds on perplexity obtained while varying the structure of the inference network. The top table depicts results for $M = 1$ and the bottom table for $M = 100$. The values along the rows and columns depict different parameters for the inference network.

| | Dimension | 1 layer | | 2 layer | | 3 layer | |
|---|---|---|---|---|---|---|---|
| | | Train | Validate | Train | Validate | Train | Validate |
| M=1 | 100 | 337.01 | 351.26 | 357.28 | 371.37 | 390.00 | 407.15 |
| | 400 | 322.94 | 340.45 | 331.36 | 347.95 | 338.09 | 355.64 |
| | Dimension | 1 layer | | 2 layer | | 3 layer | |
| | | Train | Validate | Train | Validate | Train | Validate |
| M=100 | 100 | 317.84 | 333.64 | 318.03 | 334.54 | 318.04 | 335.92 |
| | 400 | 314.75 | 332.35 | 314.04 | 332.40 | 313.95 | 332.04 |

## APPENDIX B    NETFLIX: EMBEDDINGS FOR MOVIES

The Netflix dataset Netflix (2009) comprises movie ratings of $500,000$ users. We treat each user's ratings as a document and model the numbers ascribed to each movie (from $1-5$) as counts drawn from the multinomial distribution parameterized as in Eq. 1. We train a three-layer DLGM on the dataset, evaluate $\mathcal{J}_{\mathbf{mean}}$ with 100 samples and consider two distinct methods of evaluating the learned embeddings. We cluster the movie embeddings (using spectral clustering with cosine distance to obtain 100 clusters) and depict some of the clusters in Table 6a. We find that clusters exhibit coherent themes such as documentary films, horror and James Bond movies. Other clusters (not displayed) included multiple seasons of the same show such as Friends, WWE wrestling, and Pokemon. In Table 6b, we visualize the neighbors of some popular films. In the examples we visualize, the nearest neighbors include sequels, movies from the same franchise or, as in the case of 12 Angry Men, other dramatic classics.

To compare the effect of using a model to create embeddings versus using the raw data from a large dataset directly, we evaluated nearest neighbors of movies using a simple baseline. For a query movie, we found all users who gave the movie a rating of $3$ or above (nominally, they watched and liked the movie). Then, for all those users, we computed the mean ratings they gave to every other movie in the vocabulary and ranked them based on the mean ratings. We display the top five movies obtained using this approach in Table 6. The query words are the same as in Table 6b. For most of the queries, the difference between the two is evident and we simply end up with *popular, well-liked* movies rather than relevant movies.

| Cluster Name | Movies |
|---|---|
| Documentary Films | Nature: Antarctica, Ken Burns' America: Empire of the Air, Travel the World by Train: Africa, Deepak Chopra: The Way of the Wizard & Alchemy, The History Channel Presents: Troy: Unearthing the Legend |
| Concerts | Neil Diamond: Greatest Hits Live, Meat Loaf: Bat Out of Hell, Ricky Martin: One Night Only, Beyonce: Live at Wembley, Enigma: MCMXC A.D, Sarah Brightman: In Concert |
| Horror Movies | Halloween 5: The Revenge of Michael Myers, Halloween: H2O, Creepshow, Children of the Corn, Poltergeist, Friday the 13th: Part 3, The Omen, Cujo |
| James Bond | For Your Eyes Only, Goldfinger, The Living Daylights, Thunderball, From Russia With Love, Dr. No |
| Hindi Movies | Seeta Aur Geeta, Gupt, Mann, Jeans, Coolie No.1, Mission Kashmir, Rangeela, Baazigar, Daud, Zakhm |

(a) **Clustering Movie Embeddings:** We display some of the clusters found from clustering the movie embeddings. The names were assigned based on salient features of movies in the cluster

| Cluster Name | Movies |
|---|---|
| Superman II | Superman: The Movie, Superman III, Superman IV: The Quest for Peace, RoboCop, Batman Returns |
| Casablanca | Citizen Kane, The Treasure of the Sierra Madre, Working with Orson Welles, The Millionairess, Indiscretion of an American Wife, Doctor Zhivago |
| Bride of Chucky | Bride of Chucky, Leprechaun 3, Leprechaun, Wes Craven's New Nightmare, Child's Play 2: Chucky's Back |
| The Princess Bride | The Breakfast Club, Sixteen Candles, Groundhog Day, Beetlejuice, Stand by Me, Pretty in Pink |
| 12 Angry Men | To Kill a Mockingbird, Rear Window, Mr. Smith Goes to Washington, Inherit the Wind, Vertigo, The Maltese Falcon |

(b) **Movie Neighbors:** We visualize some of the closest neighbors found to movies whose title is displayed on the column on the left

| Cluster Name | Movies |
|---|---|
| Superman II | LOTR: The Two Towers, PotC: The Curse of the Black Pearl, Raiders of the Lost Ark, LOTR: The Fellowship of the Ring |
| Casablanca | To Kill a Mockingbird, The Usual Suspects, The Shawshank Redemption, Citizen Kane, The Wizard of Oz |
| Bride of Chucky | The Matrix, Independence Day, The Silence of the Lambs, PotC: The Curse of the Black Pearl, The Sixth Sense |
| The Princess Bride | The Shawshank Redemption, Forrest Gump, LOTR: The Two Towers, LOTR: The Fellowship of the Ring, PotC: The Curse of the Black Pearl |
| 12 Angry Men | LOTR: The Fellowship of the Ring, PotC: The Curse of the Black Pear, The Godfather, Forrest Gump, The Shawshank Redemption |

(c) **Movie Neighbors [Baseline]:** We visualize some of the closest neighbors to a given query movie. We using a simple baseline that rates every movie based on average scores given by all the users who liked (rating greater than three) the query movie. LOTR (Lord of the Rings), PotC (Pirates of the Caribbean)

Table 6: **Qualitative Evaluation of Movie Embeddings:** We evaluate $\mathcal{J}_{\mathbf{mean}}^{\log}$ using 100 Monte-Carlo samples to perform the evaluation in Tables 6a and 6b.

## APPENDIX C    WIKIPEDIA: EMBEDDINGS FOR WORD

Table 7 is the full version of Table 3 in the main paper. We find that the three variants of the Jacobian vectors perform comparably across the board. The vectors obtained from shallow log-linear models appear to have the edge. The evaluation on the WordSim and SCWS datasets are done by computing the Spearman rank correlation between human annotated rankings between 1 and 10 and an algorithmically derived measures of word-pair similarity. We first compute the distances between all word pairs. Our measure of similarity is obtained by subtracting the distances from the maximal distance across all word pairs.

| Models | Spearman $\rho \times 100$ | Models | Spearman $\rho \times 100$ |
|---|---|---|---|
| Huang(G) | 22.8 | Huang (S) | 58.6 |
| Huang | 71.3 | Huang (M) | 65.7 |
| Glove | 75.9 | C&W | 57 |
| C&W | 55.3 | tf-idf-S | 26.3 |
| ESA | 75 | Pruned tf-idf-S | 62.5 |
| Tiered Pruned tf-idf | 76.9 | Pruned tf-idf-M | 60.5 |
| 1-M1 $\mathcal{J}_{\text{mean}}^{\log}$ | 65.8 | 1-M1 $\mathcal{J}_{\text{mean}}^{\log}$ | 59.9 |
| 1-M1 $\mathcal{J}_{\text{mean}}^{\text{prob}}$ | 69.7 | 1-M1 $\mathcal{J}_{\text{mean}}^{\text{prob}}$ | 61.7 |
| 1-M1 $\mathcal{J}_{\text{mean}}^{\text{pot}}$ | 66.3 | 1-M1 $\mathcal{J}_{\text{mean}}^{\text{pot}}$ | 60.4 |
| 1-M100 $\mathcal{J}_{\text{mean}}^{\log}$ | 66.3 | 1-M100 $\mathcal{J}_{\text{mean}}^{\log}$ | 58.1 |
| 1-M100 $\mathcal{J}_{\text{mean}}^{\text{prob}}$ | 70.9 | 1-M100 $\mathcal{J}_{\text{mean}}^{\text{prob}}$ | 61.1 |
| 1-M100 $\mathcal{J}_{\text{mean}}^{\text{pot}}$ | 69.5 | 1-M100 $\mathcal{J}_{\text{mean}}^{\text{pot}}$ | 59.9 |
| 3-M1 $\mathcal{J}_{\text{mean}}^{\log}$ | 45 | 3-M1 $\mathcal{J}_{\text{mean}}^{\log}$ | 54.9 |
| 3-M1 $\mathcal{J}_{\text{mean}}^{\text{prob}}$ | 46.9 | 3-M1 $\mathcal{J}_{\text{mean}}^{\text{prob}}$ | 52.2 |
| 3-M1 $\mathcal{J}_{\text{mean}}^{\text{pot}}$ | 43.9 | 3-M1 $\mathcal{J}_{\text{mean}}^{\text{pot}}$ | 53.2 |
| 3-M100 $\mathcal{J}_{\text{mean}}^{\log}$ | 59.6 | 3-M100 $\mathcal{J}_{\text{mean}}^{\log}$ | 59.6 |
| 3-M100 $\mathcal{J}_{\text{mean}}^{\text{prob}}$ | 59.6 | 3-M100 $\mathcal{J}_{\text{mean}}^{\text{prob}}$ | 59.5 |
| 3-M100 $\mathcal{J}_{\text{mean}}^{\text{pot}}$ | 57.8 | 3-M100 $\mathcal{J}_{\text{mean}}^{\text{pot}}$ | 57.9 |

(a) **WordSim353:** "G" denotes the model in Huang et al. learned only with global context in the document.

(b) **SCWS:** (S) denotes a single prototype approach versus (M) that denotes a multi-prototype approach (that leverages context)

Table 7: **Semantic Similarity in Words:** The baseline results are taken from (Huang et al., 2012). C&W uses embeddings from the language model of (Collobert & Weston, 2008). Glove corresponds to embeddings by (Pennington et al., 2014). Three words: 'y2k, insufflate, sincere' from the evaluation datasets were not found in our vocabulary after pre-processing and not used in the evaluation

## APPENDIX D  EHR DATA: EMBEDDINGS FOR DIAGNOSIS CODES

For EHR data in particular, the bag-of-diagnosis-codes assumption we make is a crude one since (1) we assume the temporal nature of the patient data is irrelevant, and (2) combining patient statistics over time renders it difficult for the generative model to disambiguate the correlations between codes that correspond to multiple diseases a patient may suffer from. Despite this, it is interesting that the Jacobian vectors still capture much of the meaningful structure among the diagnosis codes (c.f Table 2c, 8b). Here we provide additional details surrounding the evaluating of medical embeddings.

**MRM$_{\text{CCS}}$**$(V, G)$: The Agency for Healthcare Research and Quality's clinical classification software (CCS) collapses the hierarchical ICD9 diagnosis codes into clinically meaningful categories. The evaluation on CCS checks whether the nearest neighbors of a disease include other diseases related to it (if they are in the same category in the CCS). Using the ICD9 hierarchy, the authors further split the evaluation task into predicting neighbors of fine-grained and coarse grained diagnosis codes.

For a choice of granularity $G \in \{\text{fine,coarse}\}$, $V(G) \in V$ denotes the subset of ICD9 codes in the vocabulary. $\mathbb{I}_G(v(i))$ is one if the $v$'s $i$'th nearest neighbor: $v(i)$ is in the same group as $v$ according to $G$.

$$\text{MRM}_{\text{CCS}}(V, G) = \frac{1}{|V(G)|} \sum_{v \in V(G)} \sum_{k=1}^{40} \frac{\mathbb{I}_G(v(i))}{\log_2(i+1)} \qquad (6)$$

**MRM$_{\text{NDF-RT}}$**$(V, R)$: The other evaluation uses the National Drug File Reference Terminology (NDF-RT) to evaluate analogical reasoning. The NDF-RT provides two kinds of relationships ($R$) between drugs and diseases: May-Treat (if the drug may be used to treat the disease) and May-Prevent. Given $\phi_A$ as the embedding for a code $A$, this test automates the evaluation of analogies such as $\underbrace{\phi_{\text{Diabetes}}}_{r} \approx \underbrace{\phi_{\text{Metformin}}}_{v} - (\underbrace{\phi_{\text{Lung Cancer}} - \phi_{\text{Tarceva}}}_{s})$. Here $v$ is the query code and $s$ is a representation of the relationship we seek. (Metformin is a diabetic drug and Tarceva is used in the treatment of lung cancer.) The evaluation we perform reports a number proportional to the number of times

the neighborhood of $v - s$ contains $r$ for the best value of $s$ (computed from the set of all valid drug-disease relationships in the datasets.)

Given $V^* \in V$ (concepts for which NDF-RT has at-least one substance with the given relation), $\mathbb{I}_R\left(\cup_{i=1}^{40}(v - s)(i)\right)$ is one if any of the medical concepts in the top-40 neighborhood of the selected medical concept $v$ satisfies relation $R$.

$$\text{MRM}_{\text{NDF-RT}}(V, R) = \frac{1}{|V^*|} \sum_{v \in V^*} \mathbb{I}_R\left(\cup_{i=1}^{40}(v - s)(i)\right) \tag{7}$$

In both cases the choice of 40 was adopted to maintain consistency with (Choi et al., 2016). The evaluation is conducted by taking the average result over all possible seeds $s$ and the best possible seed $s$ for a query.

Table 8a depicts two examples of using the learned embeddings in the Jacobian matrix to answer tasks queries related to drug-disease pairs. Table 8b depicts clusters found in medical diagnosis codes. Table 9 is the extended version of Table 4 in the main paper (where, for the comparison on NDF-RT, we depict the results obtained from the best choice of seed $s$).

| Code 1 | Code 2 | Code 3 | Neighbors of Result |
|---|---|---|---|
| Evoxac | Sicca Syndrome | Methotrexate | Rheumatoid Arthritis |
| Biliary Atresia | Kidney Transplant | Leg Varicosity w/ Inflammation | Ligation of angioaccess arteriovenous fistula |

(a) **Medical Analogies:** We can perform analogical reasoning with embeddings of medical codes. If we know a drug used to treat a disease, we can use their relationship in vector space to find unknown drugs associated with a different disease. The queries are of the form Code 1→Code 2 ⟹ Code 3→?. Sicca syndrome or Sjogren's disease is an immune disease treated with Evoxac and Methotrexate is commonly used to treat Rheumatoid Arthiritis. " Leg Varicosity" denotes the presence of swollen veins under the skin. "Ligation of angioaccess arteriovenous fistula" denotes the tying of passage between an artery and a vein.

(b) **Medical Embeddings (Clustering):** We visualize some topical clusters of diagnosis codes.

| Label | Diagnosis Codes |
|---|---|
| Thrombosis | Hx Venous Thrombosis, Compression Of Vein, Renal Vein Thrombosis |
| Occular Atrophy | Optic Atrophy, Retina Layer Separation, Chronic Endophthalmitis |
| Drug Use | Opioid Dependence, Alcohol Abuse-Continuous, Hallucinogen Dep |

Table 9: **Medical Relatedness Measure:** Evaluated using the NDF-RT and CCS ontologies. For the evaluation on NDF-RT, there is a choice of $s$ (Eq 7) to be made. The results are reported both by averaging across all pairs of drugs-diseases used to form $s$ (avg-seed) and the best drug-disease pair (max-seed). SCUIs corresponds to the application of the method developed by (Choi et al., 2016) on data released by (Finlayson et al., 2014)

| Models | MRM$_{\text{NDF-RT}}$ (May Treat) | MRM$_{\text{NDF-RT}}$ (May Prevent) | MRM$_{\text{CCS}}$ (Fine Grained) | MRM$_{\text{CCS}}$ (Coarse Grained) |
|---|---|---|---|---|
| (De Vine et al., 2014) | 31.34/53.21 | 34.47/57.14 | 22.63 | 24.56 |
| (Choi et al., 2016) | 36.62/59.40 | 28.02/55.71 | 44.80 | 47.43 |
| SCUI | 34.89/52.75 | 30.95/48.57 | 34.16 | 37.31 |
| 1-M1 $\mathcal{J}_{\text{mean}}^{\text{log}}$ | 4.08/54.36 | 19.90/34.29 | 30.82 | 34.04 |
| 1-M1 $\mathcal{J}_{\text{mean}}^{\text{prob}}$ | 33.87/56.19 | 22.41/34.29 | 31.76 | 35.07 |
| 1-M1 $\mathcal{J}_{\text{mean}}^{\text{pot}}$ | 27.45/59.63 | 15.02/32.86 | 31.58 | 33.88 |
| 1-M100 $\mathcal{J}_{\text{mean}}^{\text{log}}$ | 33.32/54.82 | 19.02/35.71 | 33.04 | 35.78 |
| 1-M100 $\mathcal{J}_{\text{mean}}^{\text{prob}}$ | 30.70/53.90 | 21.66 / 34.29 | 32.86 | 35.66 |
| 1-M100 $\mathcal{J}_{\text{mean}}^{\text{pot}}$ | 28.49/55.73 | 15.56/31.43 | 32.87 | 35.09 |
| 3-M1 $\mathcal{J}_{\text{mean}}^{\text{log}}$ | 33.21/52.29 | 23.84/42.86 | 32.80 | 37.58 |
| 3-M1 $\mathcal{J}_{\text{mean}}^{\text{prob}}$ | 12.12/30.28 | 7.62 / 17.14 | 23.42 | 26.85 |
| 3-M1 $\mathcal{J}_{\text{mean}}^{\text{pot}}$ | 33.30/51.61 | 23.47/41.43 | 33.02 | 37.84 |
| 3-M100 $\mathcal{J}_{\text{mean}}^{\text{log}}$ | 37.00/61.47 | 23.70/42.86 | 37.54 | 40.52 |
| 3-M100 $\mathcal{J}_{\text{mean}}^{\text{prob}}$ | 9.79/21.33 | 4.39 / 11.43 | 7.82 | 8.60 |
| 3-M100 $\mathcal{J}_{\text{mean}}^{\text{pot}}$ | 36.11/60.32 | 22.26/38.57 | 37.77 | 40.87 |

