# Peer review of "Inference and Introspection in Deep Generative Models of Sparse Data"

_ICLR 2017 — rejected_

[Public Comment · Rahul Krishnan · 14 Dec 2016]
**Responses to initial comments**

[R1] Yes. Though our point was to highlight that instead of just conditioning on the input directly, thinking about how to incorporate other characteristics of the data the inference network typically has no access to (such as global first order statistics, which in the case of non-negative data corresponds to tf-idf features) helps learning.  We have updated the algorithm box to make clear the role of the additional optimization of the local variational parameters in the revised version. 

[R4] The current implementation is about 15x slower (we have emphasized this in the caption for Fig. 2) since we do 100 steps of optimizing the variational parameter. It is possible that this additional optimization is primarily helpful in the first few epochs after which one could resume training normally, though we did not experiment with such variants. 

[R3] x is the same as x_d. We have clarified the notation in the revised version. We haven’t tried our embeddings on prediction tasks for text as yet.

[Official Review · AnonReviewer1 · rating 5 · confidence 4 · 15 Dec 2016]
**Weak reject**

This paper presents a small trick to improve the model quality of variational autoencoders (further optimizing the ELBO while initializing it from the predictions of the q network, instead of just using those directly) and the idea of using Jacobian vectors to replace simple embeddings when interpreting variational autoencoders.

The idea of the Jacobian as a natural replacement for embeddings is interesting, as it does seem to cleanly generalize the notion of embeddings from linear models. It'd be interesting to see comparisons with other work seeking to provide context-specific embeddings, either by clustering or by smarter techniques (like Neelakantan et al, Efficient non-parametric estimation of multiple embeddings per word in vector space, or Chen et al A Unified Model for Word Sense Representation and Disambiguation). With the evidence provided in the experimental section of the paper it's hard to be convinced that the Jacobian of VAE-generated embeddings is substantially better at being context-sensitive than prior work.

Similarly, the idea of further optimizing the ELBO is interesting but not fully explored. It's unclear, for example, what is the tradeoff between the complexity of the q network and steps further optimizing the ELBO, in terms of compute versus accuracy.

Overall the ideas in this paper are good but I'd like to see them a little more fleshed out.

[Official Review · AnonReviewer4 · rating 5 · confidence 3 · 16 Dec 2016]
**Decent paper, but lacking novelty**

This paper introduces three tricks for training deep latent variable models on sparse discrete data:
1) tf-idf weighting
2) Iteratively optimizing variational parameters after initializing them with an inference network
3) A technique for improving the interpretability of the deep model

The first idea is sensible but rather trivial as a contribution. The second idea is also sensible, but is conceptually not novel. What is new is the finding that it works well for the dataset used in this paper.

The third idea is interesting, and seems to give qualitatively reasonable results. The quantitative semantic similarity results don’t seem that convincing, but I am not very familiar with the relevant literature and therefore cannot make a confident judgement on this issue.

[Official Review · AnonReviewer3 · rating 7 · confidence 3 · 30 Dec 2016]
**Good ; lacks more decisive experiments**

First I would like to apologize for the delay in reviewing.

Summary : In this paper a variational inference is adapted to deep generative models, showing improvement for non-negative sparse dataset. The authors offer as well a method to interpret the data through the model parameters.

The writing is generally clear. The methods seem correct. The introspection approach appears to be original. I found very interesting the experiment on the polysemic word embedding. I would however have like to see how the obtained embedding would perform with respect to other more common embeddings in solving a supervised task.

Minor :
Eq. 2: too many closing parentheses

[Official Review · AnonReviewer5 · rating 6 · confidence 4 · 03 Jan 2017]
**Interesting ideas that could have been explored in greater depth**

The paper claims improved inference for density estimation of sparse data (here text documents) using deep generative Gaussian models (variational auto-encoders), and a method for deriving word embeddings from the model's generative parameters that allows for a degree of interpretability similar to that of Bayesian generative topic models.

To discuss the contributions I will quickly review the generative story in the paper: first a K-dimensional latent representation is sampled from a multivariate Gaussian, then an MLP (with parameters \theta) predicts unnormalised potentials over a vocabulary of V words, the potentials are exponentiated and normalised to make the parameters of a multinomial from where word observations are repeatedly sampled to make a document. Here intractable inference is replaced by the VAE formulation where an inference network (with parameters \phi) independently predicts for each document the mean and variance of a normal distribution (amenable to reparameterised gradient computation).

The first, and rather trivial, contribution is to use tf-idf features to inject first order statistics (a global information) into local observations. The authors claim that this is particularly helpful in the case of sparse data such as text.

The second contribution is more interesting. In optimising generative parameters (\theta) and variational parameters (\phi), the authors turn to a treatment which is reminiscent of the original SVI procedure. That is, they see the variational parameters \phi as *global* variational parameters, and the predicted mean \mu(x) and covariance \Sigma(x) of each observation x are treated as *local* variational parameters. In the original VAE, local parameters are not directly optimised, instead they are indirectly optimised via optimisation of the global parameters utilised in their prediction (shared MLP parameters). Here, local parameters are optimised holding generative parameters fixed (line 3 of Algorithm 1). The optimised local parameters are then used in the gradient step of the generative parameters (line 4 of Algorithm 1). Finally, global variational parameters are also updated (line 5). 
Whereas indeed other authors have proposed to optimise local parameters, I think that deriving this procedure from the more familiar SVI makes the contribution less of a trick and easier to relate to.

Some things aren't entirely clear to me. I think it would have been nice if the authors had shown the functional form of the gradient used in step 3 of Algorithm 1. The gradient step for global variational parameters (line 5 of Algorithm 1) uses the very first prediction of local parameters (thus ignoring the optimisation in step 3), this is unclear to me. Perhaps I am missing a fundamental reason why that has to be the case (either way, please clarify).
The authors argue that this optimisation turns out helpful to modelling sparse data because there is evidence that the generative model p_\theta(x|z) suffers from poor initialisation. Please, discuss why you expect the initialisation problem to be worse in the case of sparse data.

The final contribution is a neat procedure to derive word embeddings from the generative model parameters. These embeddings are then used to interpret what the model has learnt. Interestingly, these word embeddings are context-sensitive once that the latent variable models an entire document.

About Figures 2a and 2b: the caption says that solid lines indicate validation perplexity for M=1 (no optimisation of local parameters) and dashed lines indicate M=100 (100 iterations of optimisation of local parameters), but the legends of the figures suggest a different reading. If I interpret the figures based on the caption, then it seems that indeed deeper networks exposed to more data benefit from optimisation of local parameters. Are the authors pretty sure that in Figure 2b models with M=1 have reached a plateau (so that longer training would not allow them to catch up with M=100 curves)? As the authors explain in the caption, x-axis is not comparable on running time, thus the question.

The analysis of singular values seems like an interesting way to investigate how the model is using its capacity.  However, I can barely interpret Figures 2c and 2d, I think the authors could have walked readers through them.

As for the word embedding I am missing an evaluation on a predictive task. Also, while illustrative, Table 2b is barely reproducible. The text reads "we create a document comprising a subset of words in the the context’s Wikipedia page." which is rather vague. I wonder whether this construct needs to be carefully designed in order to get Table 2b.

In sum, I have a feeling that the inference technique and the embedding technique are both useful, but perhaps they should have been presented separately so that each could have been explored in greater depth.

[Public Comment · Rahul Krishnan · 14 Jan 2017]
**Rebuttal**

We thank all the reviewers for their careful reading of our work and for their suggestions. 

Below we highlight additional experiments carried out based on suggestions by reviewers and clarify what set out to achieve with this work. Following which, we respond to individual reviewers inline. 

A] We have added to the paper additional experiments and insights that we highlight here: 
* We study how the proposed method for learning performs when varying the sparsity and dimensionality of the data. (Figure 2(c))
* We have added a section that sheds light on how the feature representations are constructed when using gradients with respect to z in linear and non-linear models. (the last subsection of Section 3: Methodology)
* We have studied the effect of varying hyperparameters in the inference network while continuing the optimize the variational parameters (Appendix A)
* We make code for our work available here:

[Final Decision · Program Chairs · 06 Feb 2017]
**ICLR committee final decision**

The paper introduces a number of ideas / heuristics for learning and interpreting deep generative models of text (tf-idf weighting, a combination of using an inference networks with direct optimization of the variational parameters, a method for inducing context-sensitive word embeddings). Generally, the last bit is the most novel, interesting and promising one, however, I agree with the reviewers that empirical evaluation of this technique does not seem sufficient. 
 
 Positive:
 -- the ideas are sensible 
 -- the paper is reasonably well written and clear
 
 Negative
 -- most ideas are not so novel
 -- the word embedding method requires extra analysis / evaluation, comparison to other methods for producing context-sensitive embeddings, etc